# The interaction of relative age with maturation and body size in female handball talent selection

**Zsófia Tróznai**[1]*, **Katinka Utczás**[1], **Júlia Pápai**[2], **Gergely Pálinkás**[1], **Tamás Szabó**[3], **Leonidas Petridis**[1,3]

**1** Research Center for Sports Physiology, Hungarian University of Sports Science, Budapest, Hungary, **2** Independent Researcher, Balatonszabadi, Hungary, **3** Sport Sciences and Diagnostic Research Center, Hungarian Handball Federation, Budapest, Hungary

* troznai.zsofia@tf.hu

**Data Availability Statement:** All relevant data are within the manuscript and its Supporting Information files.

## Abstract

The relative age effects (RAEs) and biological maturation are two distinct factors that have been identified to affect talent identification and selection. Previous research has suggested that talent selection should include sport-specific technical tasks instead of body size and/or physical test measurements, assuming that the technical tasks are less influenced by variations in maturation. Our purpose was to examine the prevalence of RAEs and to assess biological maturity, body size, and body composition within a single talent selection program for female handball players. Team coaches' recommendations, handball-specific drills, and in-game performance were the selection criteria. Birth distribution of all U14 female handball players were analysed ($N = 3198$) grouped in two-year age cohort. Measurements of body size, body composition (InBody 720), and bone age were performed in all players who were selected to participate in the selection program ($n = 264$) (mean±sd age: 13.1±0.6 years) and in a sample of not-selected players ($n = 266$) (mean±sd age: 13.2±0.6 years). Players were grouped in quarter-year intervals based on their date of birth. Chi-square was used to examine quartile distributions, differences between quartiles were tested with one-way analysis of variance, whereas differences between the selected and not-selected groups were examined with independent sample *t*-test (Cohen *d* effect size). Binary logistic regression was used to determine the effects of the predictors on the selection. In terms of all registered players, there was no difference in birth distribution. RAEs appeared at the first selection stage and were evident at all following stages. Quartiles differed only between the first and the last quartiles in body size and muscle mass. Only bone age differed between consecutive quartile or semi-year groups. Body size, body composition, and maturity had a significant, but of moderate power, effect on the selection. Larger body height increased the likelihood of selection by about 12%, larger muscle mass by 12% to 25%, larger bone age by 350–400%, while larger percent body fat decreased selection chances by 7%. The sport-specific criteria could not eliminate the prevalence of the RAEs. Relative age was connected to bone age, but not convincingly to body size and muscle mass. Although bone age had the largest effect on the selection, this was not associated with larger body size or muscle mass. Early maturation increased selection chances mainly during the coaches' subjective

**Funding:** The author(s) received no specific funding for this work.

**Competing interests:** The authors have declared that no competing interests exist

evaluation, but not convincingly when sport-specific tasks were applied. Given that differences were mostly evident between players of more than 1.5-year variation in their chronological age, one-year age cohorts within talent selection or the rotation of the cut-off dates of the bi-annual age grouping could be a promising strategy while also including maturation status and relative age in performance evaluations.

## Introduction

To increase competitiveness in international competitions, talent identification programs are commonly adapted in youth sport aiming to select the most talented players within a single age group [1–3]. Selected players usually receive additional training and sport science support and have more competition opportunities [4] compared to non-selected players. It has been suggested, however, that talent selection is biased by the relative age effects [5–8] (RAEs) and biological maturation [9–11]. The RAEs refer to birth inequalities between players born early and late within a single selection period (e.g., calendar year) [12], whereas maturation refers to the progress towards adulthood [9]. According to earlier reports [6, 13], relative age has been associated and explained with maturation status suggesting that the advantage of the relative older athletes is often attributed to their more advanced biological maturity. However, recent evidence support that these two factors are independent from each other and constitute separate constructs within talent selection and development [14–16]. Yet, both factors may significantly influence performance potentials and consequently talent selection [14]. Accordingly, relatively younger or late maturing players are often excluded from selection, which may negatively affect their long-term athletic career [17, 18].

In handball, body size and the physical qualities (e.g. speed, strength, power) are considered important factors for successful performance [19–22]. Such demands, however, may favour early maturing players considering the positive association of advanced maturation with anthropometric features and physical profile [23, 24]. In youth handball players, for example, early maturing players exhibited larger body dimensions and outperformed late maturing players in physical fitness tests [25–27]. Interestingly, the influence of early maturation on performance metrics seems to be more evident in boys, but not convincingly clear in girls [16, 25].

In addition to maturation, relative age may also impact performance potentials and competitive success, although to a lesser degree than maturation according to a study with elite soccer players [28]. With regard to the RAEs within handball research, diversified results have been reported in the literature. Significant RAEs have been found within talent selection systems in Norway [29] and Spain [30] suggesting unequal distribution among selected players in favour of relative older players. Relative age has been also related to game performance indicators with relatively older players playing more minutes compared to relatively younger players [29, 31, 32]. However, in a study including all participating teams in Tokyo 2020 Olympics [33], there was no RAE among female players neither when examined by playing position nor by world region. Similarly, no RAE was found among senior players who had been selected to participate in international competitions for the senior Norwegian international team [29].

Selection bias, particularly in team sports, may be further magnified when selection criteria are based primarily on measurements of body dimensions and/or physical tasks. For instance, in talent selection among handball players anthropometric and fitness test measures were included for talent identification purposes in young male [34–36] or in female players [36]. To

reduce the magnitude of maturational bias in talent development programs, Cobley et al. [6] recommended that selection should include mainly technical and movement coordination skills. Likewise, in a study with handball players, Matthys et al. [24] reported that sport-specific tasks are less influenced by body size and maturation than physical tests, thus their inclusion in talent selection could potentially separate talent identification from variations in maturity status.

Despite the available literature in talent development research, most findings rely on data from male athletes. A gender data gap seems to exist in this field with proportionally less data on female athletes compared to male athletes [37]. Further, it seems that the relevance of body dimensions and physical characteristics in game performance is not similar in men's and women's sport [38]. Given the continuously growing popularity of female sports, it is important to understand how the interaction of relative age with maturation and body size may affect talent development and the pathway across elite sports in a cohort of only female athletes separately from male athletes. Such data could improve the quality and efficiency of talent development systems designed more specifically for female athletes.

Within a single talent selection program based on coaches' recommendations and sport-specific tasks, the purpose of this study was to assess maturity status, body size, and body composition in relation to relative age among selected and not-selected female handball players. Since the talent selection program consisted of technical drills, a second aim was to examine the association of the three factors (maturation, body size, and body composition) with the outcome of the selection. Based on the generally accepted assumption, we expected that relatively older players would be more advanced in their biological development and would exhibit larger body dimensions and muscle mass compared to the relatively younger players, which would benefit them in the selection.

## Material and methods

### Experimental design

Using an observational, case-control design, this study examined relative age along with body dimensions and biological maturity within a single, multi-level talent selection program for female handball players. First, relative age of all female handball players born in 2008–2009 ($N = 3198$) was analysed using the corresponding average Hungarian population as control data ($N = 95203$). Then, we compared birth distribution at every stage of the selection program to that of all registered players to examine the influence of the selection program on the RAEs. Birth data of the handball players and the corresponding population were obtained after permission from the Hungarian Handball Federation and from the Hungarian Central Statistical Office, respectively. To examine the potential effects of body size, body composition, and biological maturity on every stage of the selection process, we performed measurements in all selected players and in a sample of not-selected players, who served as controls. Grouping criterion was the outcome of the selection process (selected vs. not-selected). Measurements were completed from November 1, 2021 until March 31, 2022. The study was approved by the Research Ethics Committee of the Hungarian University of Sport Science (Approval number: TE-KEB/23/2021).

### Female handball in Hungary

Women's handball is one of the most popular and most successful team sports in Hungary. About 35.000 senior and age-group female athletes are registered players according to the records of the Hungarian Handball Federation [39]. The domestic governing body for handball is the Hungarian Handball Federation (HHF), member of the European Handball Federation.

Responsibilities of the HHF include administration and organization of all senior and age-group competitions, funding and management of the national teams, support and development of talented players, and also coaches' education and accreditation. Competitive handball is organized within clubs and academies with official competitions beginning from the age of 10 years. Competitions are organized in one-year intervals from U10 to U15 age-groups and in two-year intervals from U16 to U20 age-groups. To participate in official competitions, clubs must be recognized members of the HHF. Handball in Schools Program was launched in 2013 by the HHF for children of 8–11 years offering training and competition opportunities organized in regional tournaments and festivals. Currently, there are three progressive levels in coaching education: basic level, intermediate level, and higher level. Coaching practice requires a valid coaching license, depending on the age-group and competition level.

## The selection program

The selection program consisted of four selection stages, starting from the local (club), basic stage to the national stage. An overall illustration of the selection program is presented in Fig 1.

The selection stages were:

Basic stage (1): coaches and staff members out of registered players of all handball clubs and teams recommended players to participate in the Selection Program.

County stage (2): selected players from the basic stage participated at this selection stage, which included handball-specific generic skills, position-specific technical drills, and in-game performance. Detailed description of the selection tasks has been reported elsewhere [40]. Briefly, the handball-specific skills consisted of two tasks, the same for all players, one for defensive and one for offensive skills. For the defensive task, the players performed defensive footwork as quickly as they can between 9 cones placed in a zigzag line. For the offensive task, players had to perform a dribbling-shooting task two times consecutively, first dribbling the ball in a straight line and then in a zigzag line ending both times with a jump shot on goal. Points were awarded for the time results for the footwork task (maximum 5 points) and for the time duration (4 points) and the success of the shots (1 point) for the dribbling–shooting task. Both tasks were repeated two times with a short rest in-between. Position-specific drills included technical tasks according to playing positions: two tasks for backcourts (maximum 10–10 points), two for the wings (maximum 10–10 points), one for the pivots (maximum 20 points), and one for goalkeepers (maximum 20 points). Selection coaches evaluated passing and shooting accuracy, goal scoring, and technical execution. In-game performance evaluation

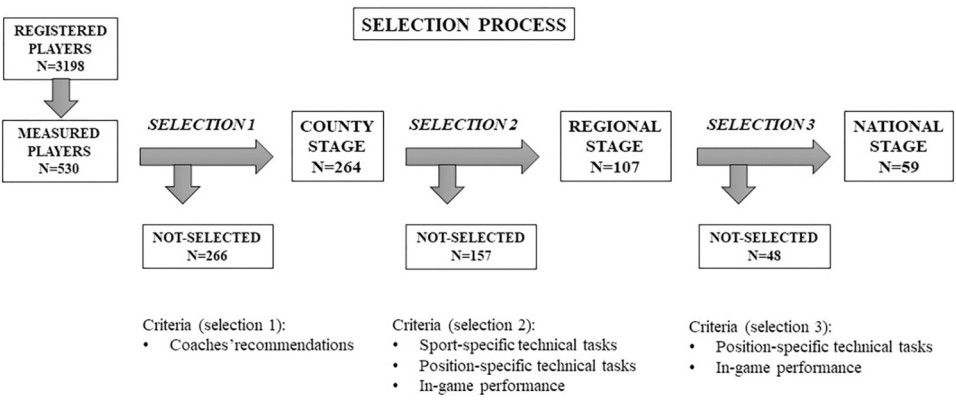

**Fig 1. Flowchart of the selection program.**

included technique and efficacy of offensive and defensive movements (5–5 points, respectively). The players with the highest scores were selected for the next (regional) level.

Regional stage (3): Players performed only the position-specific tasks, but under time pressure. At this stage there was no rest between attempts; the tasks had to be performed at a higher intensity, thereby, increasing the level of difficulty. Evaluation and scoring were the same as at county stage. In-game performance was again evaluated using the same criteria and scoring as at county stage. Players with the highest scores were selected for the next (national) level.

National stage (4): At this stage, the evaluation included only in-game performance aiming to select the players for the age group national team. The selection for the national team was not examined in this study.

## Participants

Measurements were performed in a sample of $n$ = 530 players (birth date from 1 January 2008 to 31 December 2009). The players had to be registered members of handball clubs holding a valid license to participate in official regional or national competitions. The sample consisted of $n$ = 264 selected players (mean±sd age: 13.1±0.6 years), who were selected to participate in the National Selection Program (selection being inclusion criteria) and of a subsample ($n$ = 266) of not-selected players (mean±sd age: 13.2 ±0.6 years). Covering all competition levels and regions a simple random sampling was applied to recruit not-selected players. The cohort (selected and not-selected players) represented ~17% of the total handball players in this bi-annual age group. Before the tests, the players and their parents received written information about the type and risk of the measurements and then the parents/guardians gave written consent for their children to participate. Consent was required by the Institutional Ethics Committee to provide the ethical approval for the study. We divided the players into eight groups based on their date of birth in quarter-year intervals (from Q1 to Q8). Frequencies of the quartile groups are presented in Table 1.

## Anthropometry, body composition, and biological status

Prior to performing the selection tasks, the players took part in anthropometric, body composition, and bone age measurements. Due to injury or technical limitations, the bone age of 21 players and the body composition of 2 players was not measured. Anthropometric measurements were taken based on the recommendations of the International Biological Program

**Table 1. Frequencies by relative age group across all measured stages.**

| Birth year | | Average population | Basic stage | | County stage | | Regional stage | | National stage | |
|---|---|---|---|---|---|---|---|---|---|---|
| | | | Registered players | Measured players | Not-selected | Selected | Not-selected | Selected | Not-selected | Selected |
| 2008 | Q1 | 11695 | 406 | 79 | 27 | 52 | 23 | 29 | 11 | 18 |
| 2008 | Q2 | 11664 | 386 | 68 | 30 | 38 | 19 | 19 | 5 | 14 |
| 2008 | Q3 | 13000 | 424 | 80 | 27 | 53 | 25 | 28 | 12 | 16 |
| 2008 | Q4 | 11967 | 351 | 53 | 24 | 29 | 21 | 8 | 4 | 4 |
| 2009 | Q5 | 11508 | 424 | 62 | 37 | 25 | 21 | 4 | 1 | 3 |
| 2009 | Q6 | 11428 | 409 | 79 | 52 | 27 | 17 | 10 | 6 | 4 |
| 2009 | Q7 | 12578 | 422 | 60 | 37 | 23 | 18 | 5 | 5 | 0 |
| 2009 | Q8 | 11363 | 376 | 49 | 32 | 17 | 13 | 4 | 4 | 0 |
| **Total** | | **95203** | **3198** | **530** | **266** | **264** | **157** | **107** | **48** | **59** |

Q1-Q8: quartiles groups two years period. (2008: Q1: Jan.-Mar.; Q2: Apr.-Jun.; Q3: Jul.-Sep.; Q4: Oct.-Dec.) (2009: Q5: Jan.-Mar.; Q6: Apr.-Jun.; Q7: Jul.-Sep.; Q8: Oct.-Dec.)

(IBP) [41]. Body height was measured with an anthropometer (DKSH Switzerland Ltd, Zurich, Switzerland) to the nearest millimetre. Test-rest reliability for body height was ICC = 0.99, 95%CI (0.998–1.000). Body mass and body composition were determined using Inbody 720 (Biospace Co., Seoul, Korea) bioimpedance device. The athletes were measured in the morning hours wearing shorts and t-shirts. Participants were instructed not to consume food or drink two hours and not to perform strenuous physical activity 24 hours before the measurements. Percent body fat and muscle mass were processed. Test-retest reliability for body mass was: ICC = 1.00, 95%CI (1.000–1.000); for skeletal muscle mass: ICC = 0.99, 95%CI (0.998–1.000); and for percent body fat: ICC = 0.99, 95%CI (0.996–1.000).

Biological maturity was estimated based on bone age using an ultrasound-based device (Sunlight BoneAge, Sunlight Medical Ltd, Tel Aviv, Israel). This method estimates the stage of skeletal development and has been found reliable in boys up to 16 and in girls up to 15 years [42]. Measurements were performed at the wrist region of the left hand. Participants placed their arm on a horizontal surface between the transducers. The transducers were aligned with the growth zone of the forearm (radius and ulna) at the junction of the distal epiphysis and diaphysis. Then, in the initial position, the transducer was attached to the forearm at a pressure of approximately 500 g and emitted 750 kHz ultrasound at the measurement site for each measurement cycle. One measurement cycle lasted approximately 20 seconds and was repeated five times. The instrument estimated bone age (in years and months) using equations based on the speed of ultrasound (SOS) and the distance between the transducers. The difference between chronological age (CA) and bone age (BA) was used to estimate the maturity status of the participants. Test-retest reliability of bone age estimation was: ICC = 0.98, 95%CI (0.942–0.992).

## Statistical analysis

Frequencies were used to present distribution by relative age group. Data from anthropometric, body compositions, and bone age measurements were checked for outliers (Tukey method) and for normal distribution (Kolmogorov-Smirnov test). Data of 29 athletes were excluded. Test-retest reliability was checked using absolute agreement 2-way mixed-effects model intraclass correlation coefficient (ICC). Differences in observed and expected distributions were tested using chi-square ($\chi^2$) test. Intergroup differences between Q1 to Q8 groups were tested with one-way analysis of variance (partial $\eta^2$ effect size) using Tukey post hoc test with Bonferroni correction for multiple comparisons or with the Kruskal-Wallis test when normality was violated. Differences between the selected and not-selected groups were examined using independent sample *t*-test (Cohen *d* effect size) or Mann-Whitney *U*-test when normality was violated. Finally, binary logistic regression was used to determine the effects of body size, body composition, and biological age (predictors) on the selection (dependent variable). The chance of selection (0 = not-selected, 1 = selected) were examined in separate regression models according to body size (body height and body mass), body composition (skeletal muscle mass and percent body fat), and biological status (bone age and maturity status). At the county stage we did not include bone age measurements in the regression analysis, because not-selected players were measured 2–4 months later than selected players potentially resulting in biased analysis. IBM SPSS 25.0 was used in statistical analysis; significance was set at $p < .05$.

## Results

Differences in birth distribution between the registered handball players and the average population were not significant ($\chi^2 = 12.6$; $p = .081$). Significant differences in relative age

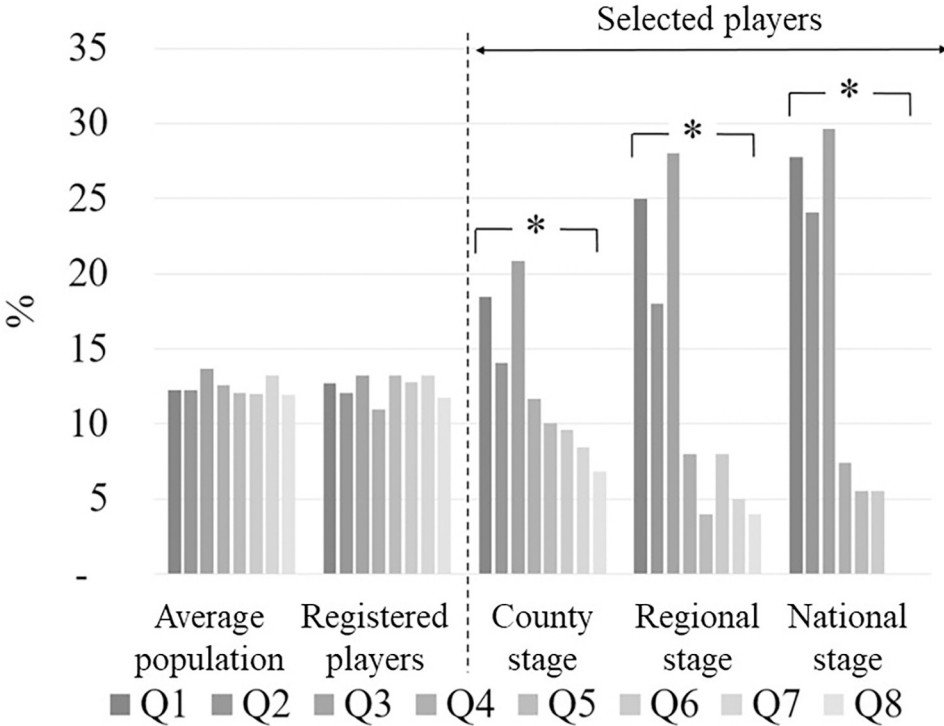

**Fig 2. Birth distribution of the average population, of all registered players in the examined age group, and of the selection program.** Q1-Q8: quartiles groups two years period (2008: Q1: Jan.-Mar.; Q2: Apr.-Jun.; Q3: Jul.-Sep.; Q4: Oct.-Dec.) (2009: Q5: Jan.-Mar.; Q6: Apr.-Jun.; Q7: Jul.-Sep.; Q8: Oct.-Dec.)*: $p < .05$ compared to registered players.

distribution were observed at all stages of selection when compared to the distribution of the registered players (county stage: $\chi^2 = 31.3$; $p < .001$, regional stage: $\chi^2 = 50.6$; $p < .001$, and national stage: $\chi^2 = 44.1$; $p < .001$). Between consecutive selection stages RAEs showed a marginally non-significant increase from county to regional stage ($\chi^2 = 13.9$; $p = .052$) and a non-significant increase from regional to national level ($\chi^2 = 6.8$; $p = .451$). At the last (national) stage there was no player born in the last two quartiles (Q7 and Q8) (Fig 2).

Kolmogorov-Smirnov tests showed that the distributions of body height, skeletal muscle mass, and maturity status were normally distributed, whereas body mass, percent body fat, and bone age deviated from normal distribution.

Boxplots of the measured variables in relation to the players' relative age are presented in Fig 3. Significantly lower values were found for players from the last quartile (Q8) compared to players from Q1, Q3, and Q4 quartiles for body mass (Fig 3A) ($H_{501(7)} = 20.55$ $p = .005$), compared to players from Q1 and Q3 for body height (Fig 3B) ($F_{(7)501} = 3.68$, $p < .001$, $\eta_p^2 = 0.05$), and compared to players from Q1 to Q5 for skeletal muscle mass (Fig 3D) ($F_{(7)499} = 4.90$, $p < .001$, $\eta_p^2 = 0.065$). Biological maturity was lower only in the Q5 and Q6 quartiles compared to the Q8 group (Fig 3F) ($F_{(7)481} = 3.40$, $p = .001$, $\eta_p^2 = 0.048$). Percent body fat did not differ between relative age groups (Fig 3C). Only bone age differed between consecutive quartile or semi-year groups ($H_{(7)481} = 144.36$ $p < .001$) (Fig 3E).

Fig 4 summarizes the results for the examined variables between selected and not-selected players at each selection stage. Differences were evident at the first and second selection stages, however effect size of at least medium magnitude (Cohen $d > 0.5$) were found only for maturation status at first stage ($d = 0.51$, 95% CI: 0.33–0.64), and for body height and skeletal muscle

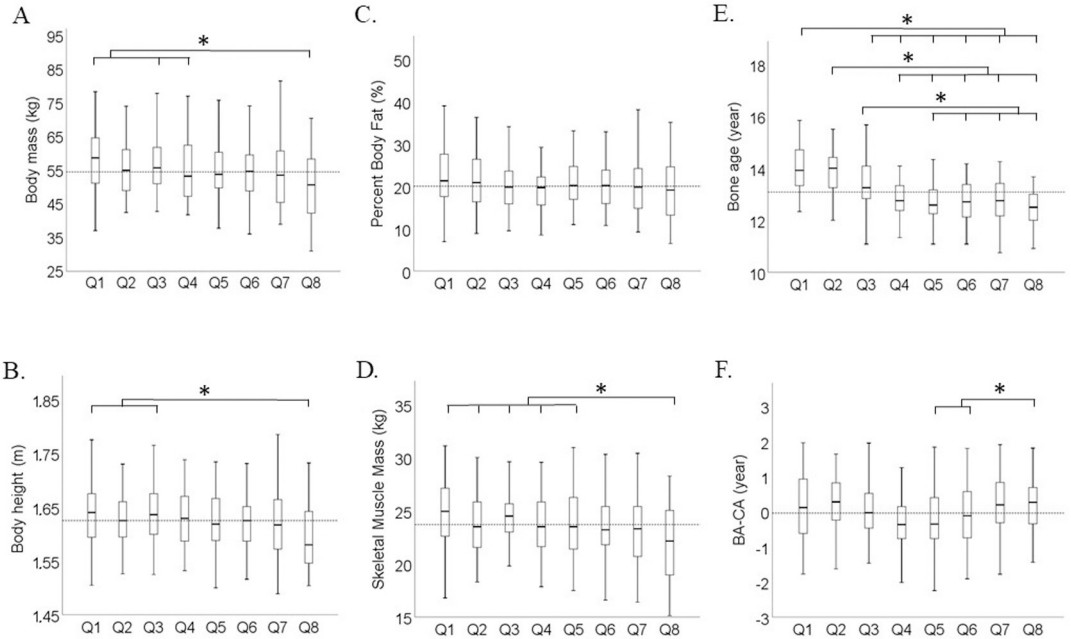

**Fig 3.** Boxplots of all measured players by relative age group for (A) body mass, (B) body height, (C) percent body fat, (D) muscle mass, (E) bone age and (F) maturity status (Bone age-Chronological age). The dashed line represents overall mean value. Q1-Q8: quartiles groups two years period. (2008: Q1: Jan.-Mar.; Q2: Apr.-Jun.; Q3: Jul.-Sep.; Q4: Oct.-Dec.) (2009: Q5: Jan.-Mar.; Q6: Apr.-Jun.; Q7: Jul.-Sep.; Q8: Oct.-Dec.)*: $p < .05$ between quartiles.

mass at the second stage ($d = 0.58$, 95% CI: 0.33–0.84 and $d = 0.56$, 95% CI: 0.30–0.82, respectively). Selected and not-selected players did not differ in any of the examined variable at the third selection stage.

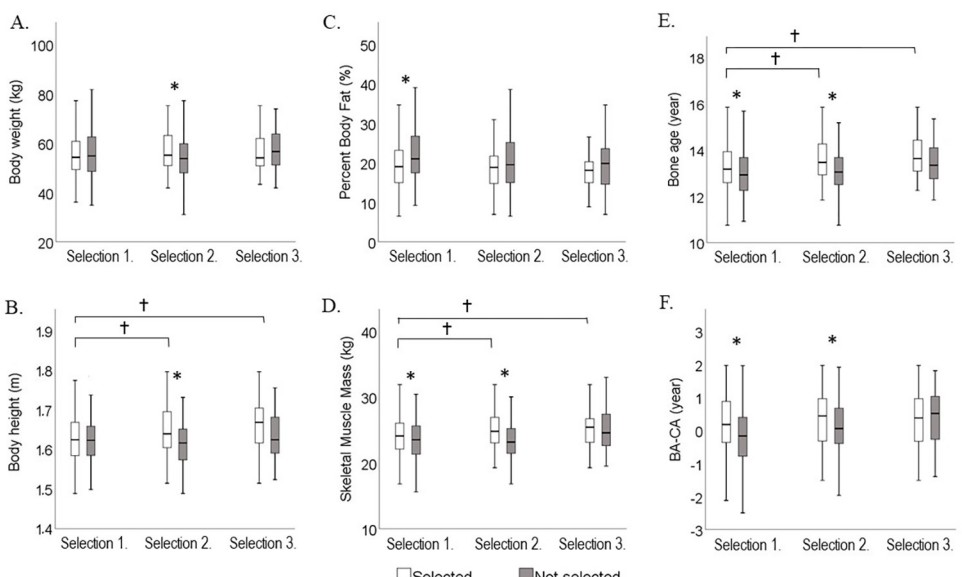

**Fig 4.** Boxplots of the selected and not-selected players at each selection stages for (A) body mass, (B) body height, (C) percent body fat, (D) muscle mass, (E) bone age and (F) maturity status (Bone age-Chronological age). *: $p < .05$ between selected and not-selected players, †:$p < .05$ between selected players at different selection stages.

**Table 2. The results of the binary logistic regression at first selection stage (dependent variable: Selected/not-selected).**

| Models | | Variables | B | Wald | p | Expl(B) | 95%CI |
|---|---|---|---|---|---|---|---|
| $\chi^2$ | 2.95 | Body mass | -0.02 | 1.95 | .162 | 0.98 | 0.96–1.01 |
| p | .229 | Body height | 0.03 | 2.59 | .107 | 1.03 | 0.99–1.07 |
| $R^2$ | 0.01 | Constant | -3.99 | 2.15 | .142 | 0.02 | |
| $\chi^2$ | 29.43 | Skeletal muscle mass | 0.12 | 12.86 | < .001 | 1.12 | 1.05–1.19 |
| p | < .001 | Percent body fat | -0.07 | 22.31 | < .001 | 0.93 | 0.90–0.96 |
| $R^2$ | 0.08 | Constant | -1.25 | 2.94 | .086 | 0.29 | |

Tables 2–4 show the results of the binary logistic regression models for body size, body composition, and biological maturity, respectively. At first selection stage, body size had no effect on the selection. Body composition components contributed significantly with an explanatory power of about 8% (Table 2). Larger skeletal muscle mass increased the likelihood of selection by about 12% for every one kilogram, while larger percent body fat decreased selection chances by 7% for every one percent of body fat.

At second selection stage all three models were significant (Table 3). Biological maturity explained about 15%, body size ***about*** 10%, and body composition about 13% of the variance in selection. Bone age had the largest effect on selection chances; one year increase in bone age increased selection chances by 350%. In addition, one centimetre increase in body height and one kilogram increase in muscle mass increased selection chances by about 12% and 25% respectively.

At third selection stage the explanatory power of the three models decreased, significant effect was found for biological maturity and body size (Table 4). One year increase in bone age increased selection chances by more than 4 times, whereas one centimetre increase in body height by about 10%.

## Discussion

The purpose of this study was to investigate the interaction of relative age with body size, body composition, and biological maturity within a single selection program among adolescent female handball players. In general, selected players were taller, had larger muscle mass, less percent body fat, and were biologically more developed than not-selected players, which is not surprising considering the importance of physical qualities in handball [19, 20, 43, 44]. However, the connection of these characteristics to the relative age was not straightforward.

Quite often talent selection system and the progression to the elite stages during an athletic career have inadvertently led to a prevalence of the RAEs in the selection process [6, 8, 45]. In

**Table 4. The results of the binary logistic regression at third selection stage (dependent variable: Selected/not-selected).**

| Models | | Variables | B | Wald | p | Expl(B) | 95%CI |
|---|---|---|---|---|---|---|---|
| $\chi^2$ | 11.34 | Bone Age | 1.43 | 9.27 | .002 | 4.19 | 1.67–10.52 |
| p | .003 | Bone age-Chronological age | -1.55 | 8.16 | .004 | 0.21 | 0.07–0.62 |
| $R^2$ | .15 | Constant | -18.8 | 9.05 | .003 | <0.01 | |
| $\chi^2$ | 6.21 | Body mass | -0.49 | 2.35 | .125 | 0.95 | 0.89–1.01 |
| p | .045 | Body height | 0.10 | 5.66 | .017 | 1.10 | 1.02–1.19 |
| $R^2$ | .08 | Constant | -12.98 | 4.95 | .026 | <0.01 | |
| $\chi^2$ | 2.53 | Skeletal muscle mass | 0.07 | 0.82 | .367 | 1.07 | 0.93–1.23 |
| p | .282 | Percent body fat | -0.05 | 1.99 | .158 | 0.95 | 0.89–1.02 |
| $R^2$ | .03 | Constant | -0.51 | 0.08 | .778 | 0.60 | |

**Table 3. The results of the binary logistic regression at second selection stage (dependent variable: Selected/not-selected).**

| Models | | Variables | B | Wald | $p$ | Expl(B) | 95%CI |
|---|---|---|---|---|---|---|---|
| $\chi^2$ | 27.23 | Bone Age | 1.25 | 20.15 | < .001 | 3.51 | 2.03–6.06 |
| $p$ | < .001 | Bone age-Chronological age | -0.80 | 6.70 | .010 | 0.45 | 0.25–0.82 |
| $R^2$ | .15 | Constant | -16.88 | 21.15 | < .001 | <0.01 | |
| $\chi^2$ | 19.85 | Body mass | -0.01 | 0.24 | .622 | 0.99 | 0.96–1.03 |
| $p$ | < .001 | Body height | 0.11 | 14.40 | < .001 | 1.12 | 1.05–1.18 |
| $R^2$ | .10 | Constant | -17.59 | 17.59 | < .001 | <0.01 | |
| $\chi^2$ | 24.63 | Skeletal muscle mass | 0.23 | 20.05 | < .001 | 1.25 | 1.14–1.38 |
| $p$ | < .001 | Percent body fat | -0.06 | 5.98 | .014 | 0.95 | 0.91–0.99 |
| $R^2$ | .13 | Constant | -4.78 | 16.20 | < .001 | 0.01 | |

addition, biological maturation seems also to influence the selection process independently from relative age [14]. To reduce the impact of maturity status, previous research [6, 46] has suggested using technical drills as selection criteria because the latter seem to be independent from biological development [24]. Applying only sport-specific criteria in the selection process, the results revealed significant RAEs at all selection stages and more advanced bone age for the selected players, particularly at the first selection stage. Considering that birth distribution among all registered players in this age group did not differ from that of the average population, the results indicate that the RAEs initiated during the selection process. These results are in accordance with previous findings in handball, which reported a selection bias in favour of the relative older players. More specifically, significant RAEs during talent development pathway were found in German handball [47] and in Danish handball [48], or among elite players participating in international competitions [32]. The magnitude of the RAEs were comparable also to that reported earlier in a selection process of similar structure [40], indicating a systemic effect of the RAEs even in this type of selection. It should be noted however, that the RAEs were evident only within the bi-annual age grouping, quartiles within one-year intervals did not differ significantly. Bi-annual age grouping is a common practice in international and domestic competitions; however, and in line with previous reports [29, 47], it seems that during the selection, the two-years age categories may increase the magnitude of the RAEs reinforcing the need to consider alternative grouping strategies.

Typically, relatively older players exhibit larger body size [49] and more advanced maturity [13] compared to the relatively younger players gaining in this way significant performance benefits. To test this hypothesis, we compared these measures according to relative age groups. The results did not convincingly confirm this assumption. Based on the entire sample (including both selected and not-selected players), differences between relative age groups were sporadic and inconsistent. In addition, significant differences were evident only between the first and the last quartile groups (e.g., in body height and muscle mass), that is, between players of above 1.5 years difference in chronological age. In line with these results, Camacho et al. [50], did not find differences in body size and percent body fat between female players born in the first and second semester of the same calendar year suggesting that relative older players are not clearly taller or more muscular compared to relative younger players. Bone age, as an indicator of biological development, was the only measure, which showed more consistent differences across relative age with relatively older players having higher values compared to their relative younger peers. Importantly, however, more advanced bone age was not definitely associated with larger body dimensions and muscle mass.

Another relevant factor is the maturity status, which is expressed as the difference between biological and chronological age. Previous research [13, 51] has suggested that among

relatively younger groups a higher proportion of early maturing players may be identified, likely compensating in this way their difficulty to be selected. Our findings however did not support this belief. The distribution of maturing status was similar between quartile groups suggesting that prior to the selection, the proportion of early/on-time/late maturing players follows a normal pattern in all relative age groups. In other words, relative younger groups did not significantly exhibit higher proportion of early maturing players. Confirming reports from soccer [4], these findings support the independent nature of relative age and maturity and together with the bone age data, indicate that the differences in biological development are simply attributed to quarter- or semi-year differences in chronological age, but not to variations in maturity status.

Interestingly, despite the differences in bone age between relatively younger and older players body dimensions did not differ. A possible reason is that even before the selection process, the body height and body mass of the players was around the 75th percentile of the normative data for the average population (1.63 m and 55 kg respectively) [52] suggesting that handball attracts girls who are genetically predisposed to large body dimensions independently of their relative age or maturation. Noteworthy, at the time of the measurements most of the examined girls were after their menarche and after their age at peak height velocity, likely indicating reduced influence of biological maturity on body size development.

A second aim was to examine the effects of anthropometric factors and maturation on the selection for each selection stage separately. Since differences between relative age groups were limited and to make the analysis more concise, we grouped the players only based on the outcome of the selection, but not according to their relative age.

At the first selection, body composition components and biological age along with maturity status appeared to favour selected players. This is an interesting finding considering that the selection at this stage was based solely on the subjective evaluation of team coaches. This confirms earlier reports suggesting that physicality and advanced maturity influence the coaches' eye even when these attributes are not selection criteria. For example, in a study with soccer players [53] coaches perceived late maturing players to have lower long-term potentials compared to early and on-time maturing players, which in turn may influence their decisions regarding players' selection and de-selection. Regarding the impact of body composition on the selection, the logistic regression results indicate a significant, but of moderate explanatory power, contribution of muscle mass and percent body fat. Larger skeletal muscle mass increased selection chances by about 12%. Interestingly, body size had no effect on the selection at this stage, as this was demonstrated by the differences between selected and not-selected players and by the logistic regression results.

The second selection included sport-specific and playing position-specific tasks. Although these tasks required mainly technical skills, the contribution of body height, muscle mass, and biological age was significant with their explanatory power being the largest at this stage (Table 3). This is confirmed also by the significant differences between selected and not-selected players (Fig 4) favouring those with larger body height and muscle mass and more advanced maturity. Sport-specific technical tasks have been suggested to be affected by biological maturation to a limited extent, thus potentially may reduce selection bias [24]. Yet, we are familiar with one study where maturity had a positive, but small, contribution to the variation in soccer-specific technical skills [54]. Our findings collectively suggest that the technique-based selection tasks cannot eliminate the impact of biological maturity on the selection among female handball players. The large selection chances of bone age in the logistic regression model (almost four times increase with increase in bone age; Table 4) clearly highlight the impact of biological development on the selection, which most probably is not limited only to the well-known benefits in body size and physical qualities, but likely affects several other key

elements in handball, like coordination, movement control, or game intelligence [55]. It has been suggested previously [13] that the selection chances of relatively younger players may increase with advanced maturity, however this was not clearly supported from our findings. Early maturation did not increase selection chances, even more it seemed to negatively affect the selection (Tables 3 and 4). A possible explanation for this discrepancy is that during puberty, advanced maturity in girls intensifies gains in body mass and subsequently in body fat mass and percent body fat [56], which, however, may impair speed, agility, and eventually athletic performance [57].

Selected players at the last selection stage represent the elite players in the examined age group accounting for less than 2% of all players in this age group. Differences between the selected and not-selected players were limited, which possibly indicates that players of different body size and body composition characteristics were excluded earlier during the selection forming in this way a more homogenous group. At this stage only bone age had a significant effect on the selection increasing selection chances. Compared to normative data from the average (non-athletic) population, the mean body height of the selected players (1.66±0.07 m) was between the 75-90th percentile [52] and it was larger by 5.6 cm than the mean value of the corresponding girls of the same age demonstrating the superior body size characteristics of the elite handball players even at a young age. On average, muscle mass of the selected players at the last stage was larger by 1.5 kg and percent body fat lower by 2.6% compared to the entire sample. Body height and body mass of the elite female handball players is comparable with that of the Greek 13.8-year-old female preliminary national handball team [19] (1.66±0.07 m; 57.3±7.8 kg) and with the selected U14 players from Croatia (1.66±0.07 m; 57.0±7.4 kg) [58].

The analysis is limited by playing positions. Talent selection was completed according to the playing position of the players, which due to the different anthropometric profile and body composition characteristics may increase variability between players (particularly between backcourts and wings) and therefore may affect the statistical analysis. It should be noted also that although the selection tasks were designed and implemented by the Handball Federation, they were based on rather empirical evidence and their validity to assess handball performance has yet to be examined. In addition, selection based first on the coaches' recommendations and then on the technical tasks may have yielded a selection bias due to the different selection criteria. Finally, and as mentioned earlier, intergroup comparisons between selected and not-selected players are presented independently from relative age quartiles. While acknowledging the relevance and importance of a separate analysis for each quartile group and due to the limited differences between quartiles, we decided to merge quartiles and highlight the differences between selected and not-selected players in a more straightforward manner.

## Conclusion

Overall, findings have demonstrated that the talent selection examined in this study (based on the coaches' subjective evaluation and sport-specific technical tasks) is influenced by the relative age effects and maturation. Relative age was connected to bone age, but not convincingly to body size and muscle mass. In addition, the small effects of the examined variables on the selection suggest a synergetic effect of maturation, body size, and muscle mass likely resulting in notable advantage during the selection.

It is important that the Hungarian Handball Federation considers relative age and maturation in the selection process particularly when evaluating performance metrics. In its present form, the only grouping factor is chronological age, which, however, creates unequal opportunities in the selection. While several intervention strategies have been suggested in the literature to counter for the influence of either RAE (e.g. age-ordered shirt numbering, minimum

quota per birth quartile, birthday banding) or maturation (e.g. maturity-based player labelling, bio-banding) [59], their efficiency in this specific selection program has yet to be examined. Education of coaches about the effects of relative age and maturation is an essential action, this could increase awareness and potentially decrease selection bias. A promising strategy would be to allow the athletes to participate in talent selection programmes more than just on one single occasion, by rotating the cut-off dates of the bi-annual grouping, so that the athletes may have the chance to experience being in the older and younger age groups (odd and even years) changing in this way expectations in skill competence and performance. Such an approach could reduce the substantial underrepresentation of players born in the second year of the bi-annual age cohort. Irrespective of the intervention strategy and as proposed by Sweeney et al. [59], an individual approach in talent selection is required, which offers maximum flexibility throughout a long-term selection process hopefully resulting in the end in more unbiased selection.

## Supporting information

**S1 File.**
(XLSX)

## Acknowledgments

We thank all the athletes, who participated in the measurements as well as their coaches and their parents/guardians for giving their consent. We are grateful for the assistance received from the Hungarian Handball Federation, which allowed this study to be carried out.

## Author Contributions

**Conceptualization:** Zsófia Tróznai, Leonidas Petridis.

**Data curation:** Zsófia Tróznai, Katinka Utczás.

**Formal analysis:** Zsófia Tróznai, Katinka Utczás, Leonidas Petridis.

**Investigation:** Zsófia Tróznai, Gergely Pálinkás, Leonidas Petridis.

**Methodology:** Gergely Pálinkás, Tamás Szabó.

**Project administration:** Tamás Szabó.

**Supervision:** Leonidas Petridis.

**Writing – original draft:** Zsófia Tróznai.

**Writing – review & editing:** Katinka Utczás, Júlia Pápai, Leonidas Petridis.

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
