## [Decision Letter · Decision Letter 0]

20 Feb 2024

PONE-D-24-03099The interaction of relative age with maturation and body size in female handball talent selection based on only sport-specific criteriaPLOS ONE

Dear Dr. Tróznai,

Thank you for submitting your manuscript to PLOS ONE. After careful consideration, we feel that it has merit but does not fully meet PLOS ONE’s publication criteria as it currently stands. Therefore, we invite you to submit a revised version of the manuscript that addresses the points raised during the review process.

 Please submit your revised manuscript by Apr 05 2024 11:59PM. If you will need more time than this to complete your revisions, please reply to this message or contact the journal office at plosone@plos.org. Please include the following items when submitting your revised manuscript:A rebuttal letter that responds to each point raised by the academic editor and reviewer(s). You should upload this letter as a separate file labeled 'Response to Reviewers'.A marked-up copy of your manuscript that highlights changes made to the original version. You should upload this as a separate file labeled 'Revised Manuscript with Track Changes'.An unmarked version of your revised paper without tracked changes. You should upload this as a separate file labeled 'Manuscript'.

We look forward to receiving your revised manuscript.

Kind regards,

Miguel Ángel Saavedra-García, Ph.D.

Academic Editor

PLOS ONE

Journal Requirements:

3. We are unable to open your Supporting Information file [Female_Talent_Selection_Database.sav]. Please kindly revise as necessary and re-upload. 

**Additional Editor Comments:**

After carefully reading the documentation sent by the reviewers. I believe that it is essential to make important changes in order to accept your paper. I recommend responding carefully to the two reviewers, as I agree with their assessments. And both reviewers request changes to your paper. As editor, I consider the article interesting and has potential for publication in PlosOne, but I understand that the reviewers have provided very interesting possibilities for improvement of your paper.

Reviewers' comments:

Reviewer's Responses to Questions

**Comments to the Author**

1. Is the manuscript technically sound, and do the data support the conclusions?

Reviewer #1: Partly

Reviewer #2: Yes

2. Has the statistical analysis been performed appropriately and rigorously? 

Reviewer #1: Yes

Reviewer #2: Yes

3. Have the authors made all data underlying the findings in their manuscript fully available?

Reviewer #1: Yes

Reviewer #2: Yes

4. Is the manuscript presented in an intelligible fashion and written in standard English?

Reviewer #1: No

Reviewer #2: Yes

5. Review Comments to the Author

Reviewer #1: I am grateful for the effort made by the authors in carrying out this scientific study, which I consider to be of relevance within the field of application of talent detection in handball.

However, I would like to express some of my concerns and suggestions in the comments made in the attached document, all in the interest of improving the quality of the manuscript. I hope they will be useful

Reviewer #2: PLOS One

Thank you for the opportunity to review your paper, that I read with great interest. The study is a novel contribution to the literature and the manuscript is clearly written. Your finding that technical tests cannot eliminate the impact of relative age and consequently biological development, is very interesting. Furthermore, that early maturation did not increase selection odds is equally intriguing. Following your reasoning, it would be interesting to use the same protocol to examine male players. I have some amendments I would urge you to consider before recommending publication.

Title

Consider to remove or revise: “…based on only sport-specific criteria” from the title.

Abstract

Be consistent in how your present the likelihood of being selected: Sometimes you use chance (percentages) and sometimes odds, making it a bit confusing to read.

Introduction

Add a paragraph that explicitly explains the relationship between relative age effects and maturation. See the work of Liam Sweeney and colleagues for inspiration.

Context

In your methodology, add a paragraph outlining the context of Hungarian women’s handball and how youth handball is organized (include information about e.g. the cultural significance of handball, number of participants, organized within clubs and/or school sport, requirements for coach certifications, and the level of professionalization in youth sports).

Discussion

Add a paragraph discussing how your findings holds practical significance for policy and practice in talent identification and selection.

Minor issues:

L124 Change “measured players” to “cohort”

L130 Please specify. I reckon this was the university ethics committee?

L133 In the methods section, you have used the local, county, and regional level as descriptors. Here, you use different terms. Please revise for consistency.

L224 See my above comment.

L315 Change “there was a study” to “we are familiar with one study”.

L339 It is unclear who “elite players” and “the entire sample” refer to here.

L353-358 Grammar off. Please revise.

6. PLOS authors have the option to publish the peer review history of their article (what does this mean?). If published, this will include your full peer review and any attached files.

Reviewer #1: No

Reviewer #2: **Yes: **Christian Thue Bjørndal

---

## [Author Response · Author response to Decision Letter 0]

12 Apr 2024

Dear Editor and Reviewers

We would like to sincerely thank the editor and the reviewers for their time to review this manuscript and for their valuable comments, which improve the manuscript’s quality. Please find our response to the comments below. Corrections in the revised manuscript appear in Track changes function.

Response to Reviewer 1

Reviewer #1: I am grateful for the effort made by the authors in carrying out this scientific study, which I consider to be of relevance within the field of application of talent detection in handball.

Dear reviewer 1. Thank you for this detailed review. 

However, I would like to express some of my concerns and suggestions in the comments made in the attached document, all in the interest of improving the quality of the manuscript. I hope they will be useful.

We revised the manuscript considering all the concerns raised in the review. Please find our responses for each comment separately below. 

ABSTRACT

 • In the first lines of the abstract, the priority areas of talent identification are mentioned. However, the authors consider putting forward the idea of implementing an integrated talent identification and development system (four corners or 360º) in which all the determining factors are included, and not one above the others. Then, of course, specify in relation to the RAE and maturity status. 

We made revisions in the abstract including also the first sentence. We tried to better address the concept that relative age and maturation are two separate factors that may influence talent selection independently from each other, as this was one of the main criticisms in the review. 

• Review the sentence on the study objective from a formal point of view as well as in relation to the comments made throughout the manuscript. 

We understand the reviewer’s point of view regarding the analysis of maturation and body dimensions for each relative age group separately, which could be more informative about the interaction of relative age with the examined factors. We have added new results in this response (lines 260-272); however (at first place) we opted not to include them in the manuscript. As mentioned in the manuscript (lines 343-345) analysis by relative age group and selection stage is quite inconsistent making the results hard to follow. Further, comparisons between selected and not-selected players could be questioned in some cases due to the few (or none) participants in several subgroups. We rephrased the purpose of the study to align more with the conducted analysis.

• Please add the average age of the players because the competition categories in international handball are of two years and the variability is large. 

Corrected accordingly for the selected and not-selected players.

• The method, in terms of procedures and statistical analysis, needs to be more precise. 

We rephrased description of the methods while also considering length limitations for the abstract. 

• The abstract should close with a powerful conclusion and, if possible, a practical application. 

We added few sentences at the end of the abstract trying to summarize the most important findings of the study. More particularly, that (1) the RAEs were evident in this type of talent selection for female players, (2) relative age was connected to bone age, but not to stature or muscle mass, and (3) early maturation increased selection chances when selection was completed based on the coaches’ subjective evaluation, but not convincingly when sport-specific tasks were applied. 

INTRODUCTION 

• Pag. 3 (line 40). The references provided are specifically related to the RAE (systematic reviews). However, the sentence also refers to a maturity status or level. Therefore, please provide some reference to this. 

Thank you for this comment, we added few references related to the influence of maturation on talent selection (line 48.) (Malina et al., 2015; Cumming et al., 2017; Hill et al. 2020)

• Malina RM, Rogol AD, Cumming SP, Coelho e Silva MJ, Figueiredo AJ. Biological maturation of youth athletes: assessment and implications. Br J Sports Med 2015; 49: 852–859. doi:10.1136/bjsports-2015-094623

• Cumming SP, Lloyd RS, Oliver JL, Eisenmann J, Malina RM. Bio-banding in sport: applications to competition, talent identification, and strength and conditioning of youth athletes. Strength Cond. J 2017;39 (2): 34–47 doi:10.1519/SSC.0000000000000281

• Hill M, Scott S, Malina RM, McGee D, Cumming SP. Relative age and maturation selection biases in academy football. J Sports Sci. 2020;38(11-12):1359-1367. doi:10.1080/02640414.2019.1649524.

• Pag. 3 (lines 35-44). In my opinion, and in relation to the previous comment, the first paragraph does not differentiate between two phenomena that have been shown to operate independently and that often tend to be confused (see Towlson, C, MacMaster, C, Parr, J & Cumming, S 2022, 'One of these things is not like the other: time to differentiate between relative age and biological maturity selection biases in soccer?', Science and Medicine in Football , vol. 6, no. 3, pp. 273-276. https://doi.org/10.1080/24733938.2021.1946133). The wording does not help to clarify both concepts and how they operate in talent detection systems. 

We have revised the main part of the introduction trying to address more clearly the independent influence of relative age and maturation on talent selection. In the revised manuscript we refer to this in (line 50-57) and also, we discuss the related literature in separate paragraphs (Ln58-64 and Ln65-72). Moreover, we have added some new references (Sweeney et al., 2023; Towlson et al., 2022; de la Rubia et al., 2024), which describe relative age and maturation as two distinct constructs within talent selection. Although it was not intentional, we understand that initially these two factors were presented as referring to the same phenomenon resulting in confusing interpretation. Hopefully, it is clearer in the revised manuscript. 

• Sweeney L, Cumming SP, MacNamara Á, Horan D. A tale of two selection biases: The independent effects of relative age and biological maturity on player selection in the Football Association of Ireland’s national talent pathway. Int J Sports Sci Coach. 2023;18(6): 1992–2003. doi:10.1177/17479541221126

• Towlson C, MacMaster C, Parr J, Cumming S. One of these things is not like the other: time to differentiate between relative age and biological maturity selection biases in soccer? Sci Med Footb. 2022;6(3): 273-276. doi: 10.1080/24733938.2021.1946133.

• de la Rubia A, Kelly AL, García-González J, Calvo JL, Mon-López D, Maroto-Izquierdo S. Biological maturity vs. relative age: Independent impact on physical performance in male and female youth handball players Biol Sport. 2024;41(3): 3–13 doi:10.5114/biolsport.2024.132999

• Pag. 3 (line 46). Reference 15 is repeated ([1]). Moreover, it and reference 16 are not useful to explain the RAE from a physical perspective. 

In the revised manuscript we removed this part and accordingly the cited papers due to the changes in the structure of the introduction. After the revisions we considered that this sentence did not contribute significantly to support the rationale of the study. 

• Pag. 3 (lines 48-50). I do not agree with this statement at all. In fact, the scientific evidence presented is limited, not considering specific handball studies by authors such as Schorer, de la Rubia, Wrang, Krahenbühl or Saavedra among others. 

We thank the reviewer for this comment. We extended handball-specific literature in the revised manuscript discriminating the papers examining the effects of relative age (Ln65-72) and papers examining maturation (Ln58-64) on either performance metrics or on talent selection.

• Pag. 3 (line 51). The term “maturity” is not correctly used. Instead, concepts such as “maturational bias” or “maturity influence” should appear. 

Corrected accordingly (changed the term maturity to maturational bias), In addition, we removed reference to the relative age from this sentence because in the cited papers (Vandendriessche et al. 2012; Matthys et al., 2012) the authors included only measures of maturation in their analysis, but not relative age. 

• Pag. 4 (lines 60-66). I agree with the authors that there is a gap in the analysis of talent identification and development processes in general, and the influence of RAE and maturity status in particular. However, there is scientific literature on some of these aspects, separately or together in a single paper:

The gap in the literature refers to the lack of data for female athletes. Most studies in the literature include male athletes and as mentioned in the manuscript we consider it important to have more data from female athletes. Such an aspect may be of interest considering also that the connection of physical characteristics and body dimensions with performance metrics and talent selection seems to be more evident in male athletes and to a lesser degree in female athletes. Thus, it is interesting to examine maturation and body size in relation to talent selection in a cohort of female athletes. We added a sentence (Lines 84-85) in the revised manuscript to address this issue and also we extended handball-specific literature including also references suggested by the Reviewer (Aouichaoui et al., 2024; de la Rubia et al., 2023).

• Pag. 4 (lines 67-72). With regard to the objective of the study, in my opinion, it should be worded differently. Neither the RAE nor the three related factors (maturation, body composition and body size) influence the chances of a player to enter the talent system in handball as this differential factor is the sport-specific tests. Moreover, if we consider the regressions performed as predictors of the selection process, the objective should be oriented in a different way by mixing both concepts. 

Indeed, the criteria of the examined selection program consisted of sport-specific tasks, which, as pointed, served as the differential factor. Exactly because of this type of selection, we aimed to examine whether typical factors (e.g. maturity, body dimensions) that often influence talent selection and performance potentials in young athletes have any effect on the selection. If the selection would have included (also) anthropometric and physical performance tests, then such analysis would be of less interest. We rephrased the purpose of the study to describe the nature and the content of the conducted analysis more precisely (Line 90-97). 

GENERAL COMMENT: 

The introduction presents a structure that does not clarify the key concepts of the research topic (RAE, maturation, talent development and identification systems, etc.). Moreover, in almost all aspects it is not specific to the sport in question (handball), providing decontextualized references in some cases. This last point would be understandable if there were no scientific literature on handball, but it is abundant in some cases. In my opinion, the Introduction section should be more precise and specific. 

In the revised introduction we tried to emphasize better the distinct influence of relative age and maturity separating the relevant literature to support the rationale of this research. In addition, we extended handball-specific literature, taking into account (1) differentiation between relative age and maturity and (2) the diversified results of the relative age effects in handball research, particularly among female players. 

MATERIAL AND METHODS 

Study Design: 

• Pag. 4 (lines 75-84). This paragraph does not provide an approach to the study design, but mixes content from the subsections Procedure and Statistical Analysis. 

We rephrased most parts of the study design specifying the type of the design that we applied in this research and adding short description of the steps we followed according to the research goals. This approach consisted of three levels: first we examined the prevalence of the RAEs at the basic level of all registered players comparing the birth distribution to that of the average population. Then, to examine the effects of the selection program on the RAEs we compared birth distribution at every stage of the selection to that of all registered players in the examined age-group. We refer to this in lines 103-107. Finally, to understand the connection of maturity, body size, and body composition with the outcome of the selection we performed measurements on all selected players and on a subsample of not-selected players. This information is mentioned in lines 107-110. We removed any reference to the statistical methods from this paragraph.

Sample: 

• Pag. 6 (lines 123-126). Explain, as exclusion criteria, the reasons why 530 players were measured out of 3198 born in that two-year cycle. 

As also mentioned above, measurements were performed on all players who were selected to participate in the selection process and on a sub-sample of not-selected players. Measurements on all 3198 players in this age group would be more than desirable, however, this seemed extremely difficult to complete. The sub-sample of the selected players was decided by coaches and the experts of the federation, who supervised and completed the selection process. We measured all selected players. The inclusion criteria in this case were the outcome of being selected. In addition, we performed measurements on a sub-sample of not-selected players. Players were recruited based on the country region and competition level for representative purposes using a simple random sampling method. Measured players represented about 17% of all registered players in the examined age group. We have added this information under methods/participants in lines 165-172.

• Pag. 7 (lines 131-132). The sample distribution into eight quartiles (odd and even year) should be stated in the text of the manuscript and not in Table 1. 

Unfortunately, we are not sure about the reviewer’s comment here. We have included in the manuscript grouping methodology based on birth quartiles (lines 173-175), however we opted to keep the presentation of each quartile’s frequency at every stage of the selection for better clarity due to the large quantity of the data. Since we generated a table containing frequency results, we considered to also include the frequency of the average population and of all registered players providing in this way a comprehensive presentation of quartile’s frequencies at every measured level and according to selected/not-selected groups. Hopefully, it can be accepted in this form. 

• Pag. 8 (lines 150-162). The protocol for measuring the three factors (anthropometric, body composition and maturity status) is very well detailed. I congratulate the authors for calculating the maturity status through skeletal age as it is one of the most reliable procedures to determine the biological development of the participants. 

Thank you. 

The Selection Program: 

• Regarding the talent programme based on only sport-specific tasks, I have three concerns: (i) given that this type of specific tasks are not carried out from phase 1 to phase 2, could it cause a selection bias? It may be that the coach's recommendation and the results of these tests are not the same so that players who are not potential talents could enter the system or vice versa according to the variables of analysis; (ii) why did the players not perform the sport-specific technical tasks in the selection process between county and regional levels; and (iii) are the tests and scores pre-validated or are they ad hoc tasks that you consider valid to assess handball performance? 

First comment: indeed, the reviewer is right that there may be a selection bias between the first and the following stages due to the different selection criteria. The structure of the selection program was designed and decided by the Hungarian Handball federation, this same structure has been implemented for the last 8 years with 8 selection programs being com

---

## [Decision Letter · Decision Letter 1]

21 May 2024

PONE-D-24-03099R1The interaction of relative age with maturation and body size in female handball talent selectionPLOS ONE

Dear Dr. Tróznai,

Thank you for submitting your manuscript to PLOS ONE. After careful consideration, we feel that it has merit but does not fully meet PLOS ONE’s publication criteria as it currently stands. Therefore, we invite you to submit a revised version of the manuscript that addresses the points raised during the review process.

As academic editor, firstly I would like to apologize for the delay, mainly due to a small health problem already overcome, and secondly for the absence of response from one of the reviewers on two occasions. Reviewer 2, who had made the decision of "Minor revision", has not replied, but, after personally reviewing his comments and the authors' replies, I understand that all the requests of reviewer 2 have been fulfilled. That is why the current decision is Minor Revision, because reviewer 1 still has to clarify some details regarding his paper.

We look forward to receiving your revised manuscript.

Kind regards,

Miguel Ángel Saavedra-García, Ph.D.

Academic Editor

PLOS ONE

Journal Requirements:

Reviewers' comments:

Reviewer's Responses to Questions

**Comments to the Author**

1. If the authors have adequately addressed your comments raised in a previous round of review and you feel that this manuscript is now acceptable for publication, you may indicate that here to bypass the “Comments to the Author” section, enter your conflict of interest statement in the “Confidential to Editor” section, and submit your "Accept" recommendation.

Reviewer #1: (No Response)

2. Is the manuscript technically sound, and do the data support the conclusions?

Reviewer #1: Yes

3. Has the statistical analysis been performed appropriately and rigorously? 

Reviewer #1: Yes

4. Have the authors made all data underlying the findings in their manuscript fully available?

Reviewer #1: Yes

5. Is the manuscript presented in an intelligible fashion and written in standard English?

Reviewer #1: Yes

6. Review Comments to the Author

Reviewer #1: (No Response)

7. PLOS authors have the option to publish the peer review history of their article (what does this mean?). If published, this will include your full peer review and any attached files.

Reviewer #1: No

---

## [Author Response · Author response to Decision Letter 1]

1 Jul 2024

Once again, we would like to thank the reviewer for the time and for the valuable comments. Please find our responses in red following all previous comments. 

 Specific and general comments (by section) 

In my opinion, the Hungarian handball talent identification system might have a weakness in this respect. Regarding the manuscript, please do not only discuss this fact in the discussion but add it as an external limitation of the study in the corresponding section after the Discussion. If you consider it, please let those responsible for the recruitment of young players in the Hungarian Handball Federation know. 

We added this thought as a limitation after the discussion under lines 407-409 (in the revised manuscript) or under lines 347-350 in the original manuscript. We are in continuous contact with the Handball Federation to share the analysis of our results and communicate our concerns and experiences. Hopefully, this may have a positive impact on the following selection programs. 

I thank you for your willingness to solve or intervene on the tests to be carried out in order to improve the selection process of young players in a country where handball is so important. However, researchers should be concerned about what to analyse and where to analyse it. Placing research in national sport talent programmes is essential to advance the science in this area, however, examining poorly designed (external) processes could reduce the quality of the study. In the case of the manuscript, please add this limitation as stated in the previous point. 

Similarly to the previous comment we also added this as limitation under lines 405-407 (in the revised manuscript) or under lines 347-350 in the original manuscript.

DISCUSSION 

The second problem of the Discussion persists, in my opinion. This section should be used to briefly state the main results, arguing the appropriate explanations for them and making them similar to/differentiating them from other results of other studies. However, in these lines (262-282) the authors only make a presentation of results already evaluated in the previous section ‘Results’. Please add to the literature. 

On the other hand, I would like to ask for more work to be done on the rebuttal letter with regard to the line numbers where the corrections are made. They do not match in any case and it is extremely difficult to check them a second time.

Thank you for this comment. We revised this paragraph trying to place our results in the literature briefly mentioning the outcomes that have been reported in previous research. The revised paragraph is now under lines 308-332 or under lines 260-285 in the original manuscript. The following citations have been added in the revised manuscript.

1 Schorer J, Wattie N, Baker JR. A New Dimension to Relative Age Effects: Constant Year Effects in German Youth Handball. PLoS ONE. 2013;8(4): e60336. https://doi.org/10.1371/journal.pone.0060336

2 Wrang CM, Rossing NN, Diernæs RM, Hansen CG, Dalgaard-Hansen C, Karbing DS. Relative Age Effect and the Re-Selection of Danish Male Handball Players for National Teams. J Hum Kinet. 2018;63(1): 33-41. doi:10.2478/hukin-2018-0004

3 Gil SM, Badiola A, Bidaurrazaga-Letona I, Zabala-Lili J, Gravina L, Santos-Concejero J, Lekue JA, Granados C. Relationship between the relative age effect and anthropometry, maturity and performance in young soccer players. J Sports Sci. 2014;32(5): 479-486. doi:10.1080/02640414.2013.832355

4 Müller L, Gehmaier J, Gonaus, C, Raschner C, Müller E. Maturity status strongly influences the relative age effect in international elite under-9 soccer. J Sport Sci Med, 2018;17(2): 216-222.

We apologize for the confusion in line numbers. After the first revision we included line numbers according to the revised manuscript and not according to the original manuscript. It is true that due to the changes in the revised manuscript, the line numbers do not match. In our current response we included the line numbers as they appear after the second revision as also according to the original manuscript for easier follow up. Hopefully, this helps.

---

## [Decision Letter · Decision Letter 2]

6 Aug 2024

PONE-D-24-03099R2The interaction of relative age with maturation and body size in female handball talent selectionPLOS ONE

Dear Dr. Tróznai,

Thank you for submitting your manuscript to PLOS ONE. After careful consideration, we feel that it has merit but does not fully meet PLOS ONE’s publication criteria as it currently stands. Therefore, we invite you to submit a revised version of the manuscript that addresses the points raised during the review process.

Some questions still remain unclear to one of the reviewers, so please be addressed to try to complete the process successfully.

We look forward to receiving your revised manuscript.

Kind regards,

Miguel Ángel Saavedra-García, Ph.D.

Academic Editor

PLOS ONE

Journal Requirements:

Reviewers' comments:

Reviewer's Responses to Questions

**Comments to the Author**

1. If the authors have adequately addressed your comments raised in a previous round of review and you feel that this manuscript is now acceptable for publication, you may indicate that here to bypass the “Comments to the Author” section, enter your conflict of interest statement in the “Confidential to Editor” section, and submit your "Accept" recommendation.

Reviewer #1: All comments have been addressed

Reviewer #2: (No Response)

2. Is the manuscript technically sound, and do the data support the conclusions?

Reviewer #1: Yes

Reviewer #2: Yes

3. Has the statistical analysis been performed appropriately and rigorously? 

Reviewer #1: Yes

Reviewer #2: Yes

4. Have the authors made all data underlying the findings in their manuscript fully available?

Reviewer #1: Yes

Reviewer #2: Yes

5. Is the manuscript presented in an intelligible fashion and written in standard English?

Reviewer #1: Yes

Reviewer #2: Yes

6. Review Comments to the Author

Reviewer #1: The authors have made a great effort to make all the modifications and changes proposed by the reviewer. I now consider the article ready for publication. Therefore, I would like to congratulate each and every one of the authors of the article.

Reviewer #2: Thank you for the opportunity to read your manuscript, which I found to be both novel and thoroughly conducted. Overall, it makes a strong contribution to the research literature. However, I find the discussion section somewhat restricted. It would benefit from more courageous argumentation and suggestions for policy and practice. It is not entirely clear how increased monitoring of RAEs and biological maturation would lead to improved talent identification and development, though it might result in fairer selections. I recommend the authors read the recent contribution by Sweeney et al. (2023) for further insights (full reference provided below).

Apart from these points, I have only a few minor issues before recommending publication. Congratulations to the authors for their efforts.

Abstract

- Lines 32-33: To improve coherence, consider using either percentages or multiplicatives consistently. Is there a specific reason for presenting the data as it is? This also applies to the results section.

Introduction

- Line 59: Change “participation” to “performance”.

- Line 72: Elaborate on the details of this study [33].

Conclusion

- Lines 418-420: This sentence is confusing. Please revise for better clarity.

References

- Sweeney, L., Taylor, J., & Macnamara, Á. (2023). Push and Pull Factors: Contextualising Biological Maturation and Relative Age in Talent Development Systems. Children, 10(1), 130.[https://doi.org/10.3390/children10010130](https://doi.org/10.3390/children10010130)

7. PLOS authors have the option to publish the peer review history of their article (what does this mean?). If published, this will include your full peer review and any attached files.

Reviewer #1: No

Reviewer #2: **Yes: **Christian Thue Bjørndal

---

## [Author Response · Author response to Decision Letter 2]

12 Sep 2024

RESPONSE TO REVIEWERS

Once again, we would like to thank the reviewer for the time and for the valuable comments. Please find our responses in red.

Review Comments to the Author

Reviewer #1: The authors have made a great effort to make all the modifications and changes proposed by the reviewer. I now consider the article ready for publication. Therefore, I would like to congratulate each and every one of the authors of the article.

 Thank you very much for your kind words.

Reviewer #2: Thank you for the opportunity to read your manuscript, which I found to be both novel and thoroughly conducted. Overall, it makes a strong contribution to the research literature. However, I find the discussion section somewhat restricted. It would benefit from more courageous argumentation and suggestions for policy and practice. It is not entirely clear how increased monitoring of RAEs and biological maturation would lead to improved talent identification and development, though it might result in fairer selections. I recommend the authors read the recent contribution by Sweeney et al. (2023) for further insights (full reference provided below).

We would like to thank the reviewer for his kind words and for the valuable comments. We revised and expanded in more details the last paragraph under conclusions where recommendations for policy are discussed taking also into consideration the paper recommended by the reviewer (Sweeney et al., 2023). As indicated in the revised paragraph, we consider it important that the selection is not limited only to one participation, but the players have the chance to participate in the talent selection process in multiple occasions by rotating the cut-off dates, so the players may experience being in the older and younger age groups potentially counteracting in this way the significant underrepresentation of players born in the second year. However, it should be noted that the efficiency of the possible intervention strategies has yet to be examined, currently there is not enough evidence to support the use of a specific intervention over another one.

Apart from these points, I have only a few minor issues before recommending publication. Congratulations to the authors for their efforts.

Thank you.

Abstract

- Lines 32-33: To improve coherence, consider using either percentages or multiplicatives consistently. Is there a specific reason for presenting the data as it is? This also applies to the results section.

We corrected multiplicative to percentages in the abstract and in the results. We found the use of percentage easier to describe the influence of the examined variables on the selection. 

Introduction

- Line 59: Change “participation” to “performance”.

corrected accordingly

- Line 72: Elaborate on the details of this study [33].

We rephrased this sentence slightly expanding on the details of the cited paper. Also, we added an additional reference (Bjørndal et al., 2018), where the authors reported no RAE among selected players at the senior age group. 

Conclusion

- Lines 418-420: This sentence is confusing. Please revise for better clarity.

We rephrased this sentence, hopefully is clearer now. 

As recommended, the following paper has been included in the revised manuscript:

Sweeney, L., Taylor, J., & Macnamara, Á. (2023). Push and Pull Factors: Contextualising Biological Maturation and Relative Age in Talent Development Systems. Children, 10(1), 130.[https://doi.org/10.3390/children10010130](https://doi.org/10.3390/children10010130)

---

## [Decision Letter · Decision Letter 3]

2 Oct 2024

The interaction of relative age with maturation and body size in female handball talent selection

PONE-D-24-03099R3

Dear Dr. Tróznai,

We’re pleased to inform you that your manuscript has been judged scientifically suitable for publication and will be formally accepted for publication once it meets all outstanding technical requirements.

Kind regards,

Miguel Ángel Saavedra-García, Ph.D.

Academic Editor

PLOS ONE

Additional Editor Comments (optional):

Reviewers' comments:

Reviewer's Responses to Questions

**Comments to the Author**

1. If the authors have adequately addressed your comments raised in a previous round of review and you feel that this manuscript is now acceptable for publication, you may indicate that here to bypass the “Comments to the Author” section, enter your conflict of interest statement in the “Confidential to Editor” section, and submit your "Accept" recommendation.

Reviewer #3: All comments have been addressed

2. Is the manuscript technically sound, and do the data support the conclusions?

Reviewer #3: Yes

3. Has the statistical analysis been performed appropriately and rigorously? 

Reviewer #3: Yes

4. Have the authors made all data underlying the findings in their manuscript fully available?

Reviewer #3: Yes

5. Is the manuscript presented in an intelligible fashion and written in standard English?

Reviewer #3: Yes

6. Review Comments to the Author

Reviewer #3: The authors have corrected everything that has been asked of them and that you are not going to add anything else, so you understand that, although everything can always be improved, the reviewers' work is already done and the paper can be accepted.

7. PLOS authors have the option to publish the peer review history of their article (what does this mean?). If published, this will include your full peer review and any attached files.

Reviewer #3: **Yes: **Helena Vila

---

## [Editor Report · Acceptance letter]

9 Oct 2024

PONE-D-24-03099R3 

PLOS ONE

Dear Dr. Tróznai, 

I'm pleased to inform you that your manuscript has been deemed suitable for publication in PLOS ONE. Congratulations! Your manuscript is now being handed over to our production team.

Kind regards, 

on behalf of

Dr. Miguel Ángel Saavedra-García 

Academic Editor

PLOS ONE